# Geosynthetic Reinforced Steep Slopes: Current Technology in the United States

**Yoo-Jae Kim [1,*], Ashley Russell Kotwal [1], Bum-Yean Cho [2], James Wilde [1] and Byung Hee You [1]** 

[1]  Department of Engineering Technology, Materials Science, Engineering, and Commercialization Program, Texas State University, 601 University Drive, San Marcos, TX 78666, USA; yjk4@twc.com (A.R.K.); wjw42@txstate.edu (J.W.); by12@txstate.edu (B.H.Y.)

[2]  Department of Fire Safety Research, Korea Institute of Civil Engineering and Building Technology, 64 Ma-doro 182beon-gil, Mado-myeon, Hwaseong-si, Gyeonggi-do 18544, Korea; Choby277@kict.re.kr

*  Correspondence: yk10@txstate.edu; Tel.: +1-512-245-6309

**Abstract:** Geosynthetics is a crucial mechanism in which the earth structures can be mechanically stabilized through strength enforcing tensile reinforcement. Moreover, geosynthetic reinforcement stabilizes steep slopes through incorporating the polymeric materials, becoming one of the most cost-effective methods in not only accommodating budgetary restrictions but also alleviating space constraints. In order to explicate on the applicability and widen the understanding of geosynthetic reinforcement technology, a synthesis study was conducted on geosynthetic reinforced steep slope. This study is very important because in not only highlighting the advantages and limitations of using geosynthetic reinforcement but also in investigating the current construction and design methods with a view to determining which best practices can be employed. Furthermore, this study also identified and assessed the optimal condition of the soil, performance measures, construction specifications, design criteria, and geometry of the slope. To further concretize the understanding of these parameters or factors, two case studies were reviewed and a summary of the best practices, existing methods, and recommendations were drawn in order to inform the employment of geosynthetics in reinforcing steep slopes.

**Keywords:** reinforcement; geosynthetics; geogrid; soil; slope

## 1. Introduction

Reinforced slopes are a form of mechanically stabilized earth that incorporate planar reinforcing elements for the construction of sloped structures with inclinations less than 70°. Structures with inclinations over 70° are classified as walls [1]. For uniform fill soil, there is a limiting slope angle under which an unreinforced slope may be built safely. The limiting angle of the slope is equal to the friction angle of the soil in the case of a cohesionless and dry material. Therefore, a slope with an angle greater than the limiting slope angle is referred to as a steep slope, which requires additional forces to maintain equilibrium [2]. The purpose of steep slope construction is to solve problems in locations of restricted right-of-way and at marginal sites with difficult subsurface conditions and other environmental constraints. Since soil has limited tensile strength, similar to concrete, reinforcement must be used to overcome this weakness [3]. To improve strength and make a soil structure self-supporting, tensile reinforcing elements are placed in the soil. The reinforcing elements can also withstand bending from shear stresses, providing increased stability to steep slopes [4].

Soil reinforcement concepts and technologies originated in prehistoric times. Straw, sticks, and branches were traditionally used to improve the quality of adobe bricks, reinforce mud dwellings, and even reinforce soil for erosion control. However, modern techniques for mechanically stabilizing

soil were introduced in the 1960s. First used in France, a method known as "reinforced earth" used embedded narrow metal straps to reinforce soil [5]. In 1972, this technique was adopted in the United States by the California Division of Highways for construction of retaining walls. Many other soil reinforcement methods were researched and implemented following the first applications in the United States [6].

Geosynthetics are modern materials used to improve soil conditions by providing tensile resistance and stability. In the case of soil reinforcement, a primary application of geosynthetics involves reinforcing steep slopes. With the use of geosynthetics, the construction of reinforced steep slopes is often more affordable and technically feasible when compared to traditional construction techniques [7]. Furthermore, geosynthetic reinforcement is a cost-effective solution for stabilizing recurring slope failures and constructing new permanent embankments [8,9].

Geosynthetic reinforcement of steep slopes was initially utilized to repair failed slopes. Instead of importing soil to reconstruct the slopes, the slide debris was salvaged and reused with the addition of geosynthetic reinforcement, resulting in reduced costs. To provide stability, multiple layers of geosynthetic materials were placed in a fill slope during reconstruction. As per Figure 1, in addition to repairing failed slopes, steep slope reinforcement has also been introduced for the construction of new embankments, the widening of existing embankments, and as an alternative to retaining walls [10]. As reinforced slopes became constructed to perform as permanent structures, the structural analysis method used for near-vertical reinforced walls was adopted, which relies on analytical predictions, factors of safety, and reduction factors. Consequently, design methods for reinforced steep slopes are reasonably conservative [8]. Research efforts regarding the design and application of geosynthetics are rapidly increasing on the international level as well, including investigations in Canada, Taiwan, and the United Kingdom [11–13].

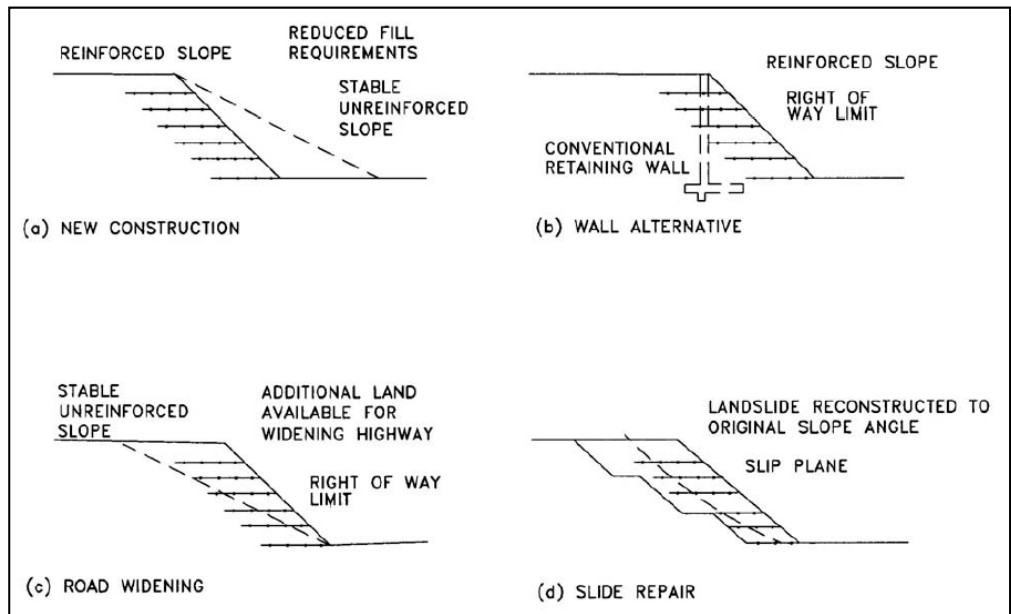

**Figure 1.** Geosynthetic Reinforced Steep Slope Applications [10].

This study provides a thorough review of the current technology on geosynthetic reinforced steep slopes. Findings from state, national, and international sources were summarized and compared to establish best practices. The study synthesized information gathered from a literature review, interviews, and surveys of transportation agencies, educational institutions, consulting engineers, manufacturers, material suppliers, and construction contractors. To obtain supplementary information and professional insight, experts were also identified from the survey.

## 2. Geosynthetic Materials

Geosynthetics are planar products manufactured from a polymeric material used with soil, rock, earth, or other geotechnical-related material as an integral part of a civil engineering project, structure, or system [14]. They are identified by the type of polymer, type of fiber or yarn, type of geosynthetic, mass per unit area or thickness, and any additional clarification needed to describe the geosynthetic.

Geotextiles are defined as any permeable textile used with foundation soil, rock, earth, or any other geotechnical engineering-related material as an integral part of a manmade project, structure, or system. Woven geotextiles are cloth-like fabrics that are formed by uniform and regular interweaving of threads or yarns in two directions. They have regular visible construction patterns and, where present, have distinct and measurable openings. Geogrids are open grid-like materials of integrally connected polymers and are stronger than most geotextiles. Their primary use is soil reinforcement, because they can withstand heavy tension loads without much deformation and have low strain compared to geotextiles. Even though different geosynthetics may be manufactured with the same base polymer, their tensile strengths can vary widely. The tensile properties of geosynthetics are affected by factors such as creep, installation damage, aging, temperature, and confining stress. Some may also be more susceptible to environmental damage, including exposure to chemicals, heat, and ultraviolet light [15,16]. The effect of long-term stress should be determined from laboratory creep tests in accordance with ASTM D5262 and extrapolated to the desired design life or carried out to rupture when possible [10,17]. In ascending order, the most creep susceptible polymers are polyester, polypropylene, and polyethylene. An increase in temperature significantly accelerates creep for polypropylene [18]. Confinement may also change the creep properties of nonwoven geotextiles; however, for woven geotextiles and geogrids, the effect of confinement on creep is negligible [19].

For soil slope applications, Alabama Department of Transportation (DOT) specifies that the geosynthetic reinforcement (either geogrid or geotextile) shall be constructed of polyester, polypropylene, or polyethylene, resistant to all naturally occurring alkaline and acidic soil conditions, and resistant to heat, ultraviolet light, and to attack by bacteria and fungi in the soil [20]. Reinforcement for soil slopes shall be any geosynthetic whose strength in the machine direction equals or exceeds the values provided in the specification. California DOT requires for the percentage of the open area for geogrids to range from 50 percent to 90 percent of the total projection of a section of the material, and geotextiles shall have an irregular or regular open area with the spacing of open areas being less than 1 cm in any direction [21]. West Virginia DOT also requires the geosynthetics to be capable of withstanding 150 hours of testing as per ASTM D4355 with no measurable reduction in the ultimate tensile strength or deterioration of the coating [22,23]. Table 1 provides additional recommendations for geosynthetic material based on soil environment [10]. Table 2 displays various geosynthetic types, properties, and test methods [1].

**Table 1.** Geosynthetic Resistance to Specific Soil Environments [10].

| Soil Environment | PET | PE | PP |
|---|---|---|---|
| Acid Sulphate Soils | NE | ETR | ETR |
| Organic Soils | NE | NE | NE |
| Saline Soils (pH < 9) | NE | NE | NE |
| Ferruginous Soils | NE | ETR | ETR |
| Calcereous Soils | ETR | NE | NE |
| Modified Soils (Lime, Cement, etc.) | ETR | NE | NE |
| Sodic Soils (pH > 9) | ETR | NE | NE |
| Soils with Transition Metals | NE | ETR | ETR |

PP = Polypropylene, PET = Polyester, PE = Polyethylene
NE = No Effect, ETR = Exposure Tests Required

**Table 2.** Geosynthetic Types, Properties, and Test Methods * [1].

| Geosynthetic Type | Weight [2] (g/m$^3$) | Ultimate [3] Tensile Strength (kN/m) | Strain at [3] Ultimate Tensile Strength (%) | Secant [3] Modulus at 10% Strain (kN/m) | Grab [4] Strength (N) | Puncture [5] Strength (N) | Burst [6] Strength (kPa) | Tear [7] Strength (N) | Equivalent [8] Darcy Permeability (m/s) |
|---|---|---|---|---|---|---|---|---|---|
| Monofilament Polypropylene Geotextile | 120–240 | 16–70 | 20–40 | 70–260 | 700–2300 | 320–700 | 2700–4800 | 200–440 | $10^{-4}$–$10^{-2}$ |
| Silt Film Geotextile | 50–170 | 12–45 | 20–40 | 50–260 | 32–1600 | 80–600 | 1400–4800 | 200–1600 | $10^{-4}$–$10^{-3}$ |
| Fibrillated Tape and Multifilament Polypropylene Geotextile | 240–760 | 35–210 | 15–40 | 175–700 | 700–6200 | 700–1100 | 4100–10400 | 440–1800 | $10^{-4}$–$10^{-3}$ |
| Multifilament Polyester Geotextile | 140–710 | 25–350 | 10–30 | 175–10500 | 700–9000 | 200–1400 | 3400–10400 | 360–2300 | $10^{-4}$–$10^{-3}$ |
| Polypropylene Geogrid | 140–240 | 8–35 | 10–20 | 90–230 | n/a | n/a | n/a | n/a | >10 |
| High Density Polyethylene Geogrid | 240–710 | 8–90 | 10–20 | 55–70 | n/a | n/a | n/a | n/a | >10 |
| Polyester Geogrid | 240–710 | 35–140 | 5–15 | 350–2600 | n/a | n/a | n/a | n/a | >10 |

[1]. The data in this table represent an average range. There may be products outside this range. No relation should be inferred between maximum and minimum limits for different tests. [2]. Method 1.1.84, Appendix B, FHWA Geotextile Engineering Manual. [3]. Wide Width Method, ASTM D-4595. [4]. ASTM D-4632. [5]. ASTM D-4833. [6]. ASTM D-3786. [7]. ASTM D-4533. [8]. ASTM D-4491. * Limited by test machine.

## 3. Foundation & Backfill Materials

To ensure successful construction of geosynthetic reinforced steep slopes, an adequate subsurface investigation should be performed for the existing foundation, as well as behind and in front of the structure to assess overall performance behavior. Foundation soil engineering considerations include the bearing resistance, global stability, settlement potential, and position of groundwater levels. The subsurface exploration program generally consists of soil soundings, borings, and test pits. The exploration must be sufficient to evaluate the geologic and subsurface profile in the area of construction. Ground improvement techniques that account for major foundation weakness and compressibility may be required to achieve adequate bearing capacity or limited total or differential settlement [4,10].

In general, select fill materials are more expensive than lower quality materials. The fill specifications depend on the application and final performance requirements of the structure. Detailed project reinforced fill specifications should be provided by the contracting agency. A high quality embankment fill meeting gradation requirements to facilitate compaction and minimize reinforcement is recommended. Table 3 presents the recommendations for granular reinforced fill [24–26].

**Table 3.** Granular Reinforced Fill Recommendations [10,24–26].

| | Sieve Size | Percent Passing (%) |
|---|---|---|
| **Gradation** **(AASHTO T-27)** | 4 in. | 100 |
| | #4 | 100–20 |
| | #40 | 60–0 |
| | #200 | 50–0 |
| **Plasticity Index** **(AASHTO T-90)** | PI (Plastic Index) ≤ 20 | |
| **Soundness** **(AASHTO T-104)** | Magnesium sulfate soundness loss less than 30% after 4 cycles, based on AASHTO T-104 or equivalent sodium sulfate soundness of less that 15% after 5 cycles. | |

West Virginia DOT specifies for all backfill material in the structure volume to be reasonably free from organic or otherwise deleterious material [22]. Prior to incorporating the soil, they require the contractor to perform one pH test in each soil type each day of operation, and the pH of the soil shall be within the allowable limits of the design for the geosynthetic material used. For most geosynthetic materials, California DOT and Florida DOT recommend a pH level for backfill material in the range of 5 to 10 [21,27]. Additionally, fill should meet the minimum required shear strength parameters as determined by direct shear or consolidated-drained triaxial tests. Pennsylvania DOT requires for the material to have a minimum angle of internal friction of 32° if no minimum shear strength parameters are indicated [28].

### 3.1. Weak Foundation Soil

Horizontal earth pressures tend to laterally spread embankments constructed on weak foundation soils. Weakened areas may be caused by sink holes, thawing ice, old stream beds, or layers of soft silt, clay, or peat. Failure will result if the foundation soil does not have adequate shear resistance to the stresses. Therefore, the design guidelines for reinforced embankments on weak soils should also consider bearing failure, rotational failure, and lateral spreading as per Figure 2 [29].

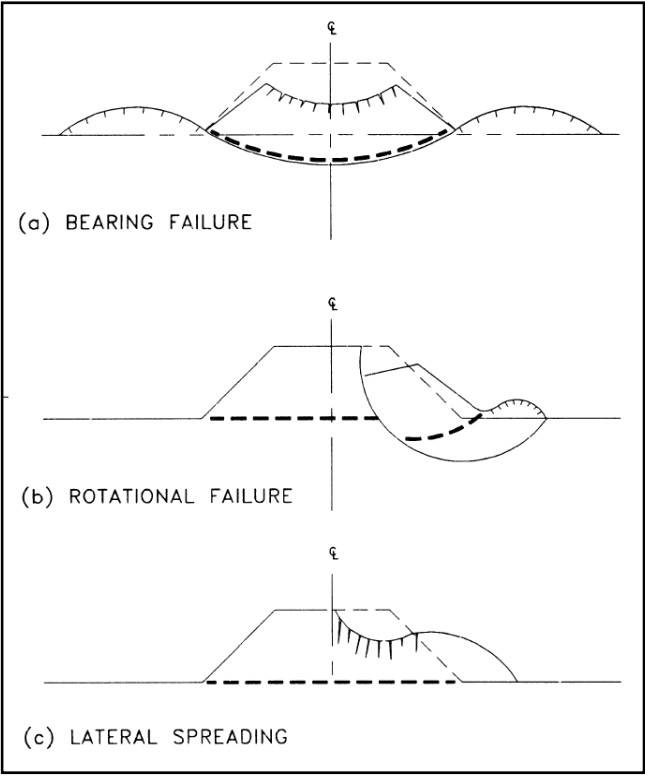

**Figure 2.** Embankment Failure Modes on Weak Foundation Soils [29].

Classical bearing capacity theory should be used when the thickness of the weak soil is much greater than the width of the embankment [30]:

$$q_{ult} = (\gamma \times H) = (c_u \times N_c) \tag{1}$$

where:

$q_{ult}$ = ultimate bearing pressure, N/m$^2$.
$\gamma$ = unit weight of the soil, N/m$^3$.
$H$ = height of the slope, m.
$c_u$ = undrained cohesion of the soil, N/m$^2$.
$N_c$ = bearing capacity factor.

If the factor of safety for bearing capacity is insufficient, the following project parameters should be considered: increasing the embankment width, flattening the slopes, adding toe berms, or improving the foundation soils. A rotational slip surface analysis can also be used to determine the critical failure surface and the factor of safety against shear instability [1]:

$$T = [(FS \times M_D) - M_R]/[R \times cos(\theta - \beta)] \tag{2}$$

where:

$T$ = tensile strength of reinforcement, N/m.
$FS$ = factor of safety, N/m$^3$/N/m$^3$.
$M_D$ = driving moment, N·m.
$M_R$ = resisting moment, N·m.
$R$ = radius, m.
$\theta$ = angle from horizontal to tangent line, degrees.

$\beta$ = 0 for brittle strain-sensitive foundation soils.

= $\theta$/2 for depth/width ratio < 0.4 and moderately compressible soils.

= $\theta$ for depth/width ratio ≥ 0.4 and highly compressible soils.

Finally, a lateral spreading or sliding wedge stability analysis should be performed. Cohesion is assumed to equal zero for extremely soft soils and low embankments. The factor of safety is calculated as follows [31]:

$$FS = (b \times tan\phi')/(K \times H) \tag{3}$$

where:

$b$ = width of the wedge, m.

$\phi'$ = effective internal friction angle of the soil, degrees.

$K$ = earth pressure coefficient.

$H$ = height of the slope, m.

The elastic modulus of geosynthetic reinforcement is relied upon to control lateral spreading over weak foundation soils. By limiting the deformation behavior of the reinforcement, excessive deformation of the embankment can be mitigated. The distribution of lateral pressure and strain is assumed to vary linearly, increasing from zero at the toe to a maximum value beneath the crest of the embankment. The maximum strain in the reinforcement should be equal to twice the average strain in the embankment. Tolerable deformation requirement (*J*) is calculated as follows:

$$J = T/\varepsilon \tag{4}$$

where:

$T$ = tensile strength of reinforcement, N/m.

$\varepsilon$ = strain limit based on type of fill soil materials (m/m).

*3.2. Marginal Backfill Soil*

Compaction difficulties and pore water pressure can create unstable conditions in slopes with marginal backfill consisting of low quality, cohesive, fine grained soil. Although granular soils are preferred due to their high strength and low pore water pressure, project budgets may require the use of marginal soils when select fill is not readily available. Design considerations include accounting for excess pore water pressure development by incorporating reinforcement-drainage composites that promote lateral drainage within the soil mass. Internal and external seepage forces must also be accounted for during the design process. It is recommended for a two-phase analysis to be performed, including a total stress analysis that ignores the reinforcement lateral drainage and an effective stress analysis that accounts for full lateral drainage. Tensile strength, pullout resistance, drainage, and filtration are the main characteristics that should be considered when selecting a geosynthetic material for marginal backfill [32].

**4. Design Methods**

The majority of survey respondents recommended the use of the FHWA guidelines for the design of reinforced slopes as per Figure 3. The technique has been adopted by many transportation agencies in the United States and South America [33]. Less advocated approaches included Eurocode and other design methods, such as EBGEO (Empfehlungen für den Entwurf und die Berechnung von Erdkörpern mit Bewehrungen aus Geokunststoffen) and British Standard 8006. Additionally, 88 percent recommended the use of geogrid or a combination of geogrid and geotextile for slope reinforcement, while others recommended independent use of geotextile. Some prefer high strength geotextiles over geogrids, as they provide a separation function and can be more cost effective in certain cases. See Appendix A for the case studies.

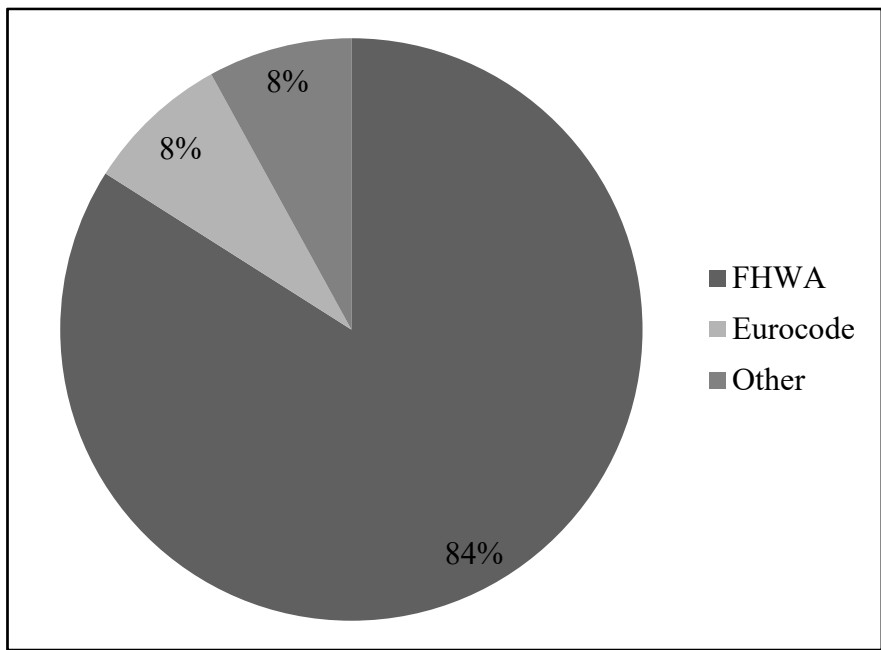

**Figure 3.** Recommended Design Methods.

The internal stability of a soil slope can be determined by four basic factors, including the slope angle in degrees ($\beta$), soil weight in N ($W$), cohesion in N/m$^2$ ($c$), and internal friction angle in phi ($\phi$) as per Figure 4. The cohesion is a measure of the forces that cement particles of soil, and the internal friction angle is a measure of the shear strength of soils due to friction. The force causing failure, the resisting strength, and the factor of safety are also dependent upon the length of the plane of weakness in m ($L$) as follows [34]:

$$\text{Force Causing Failure} = (W \times sin\beta) \tag{5}$$

$$\text{Resisting Strength} = (c \times L) + (W \times cos\beta \times tan\phi) \tag{6}$$

$$\text{Factor of Safety} = (\text{Resisting Strength/Force Causing Failure}) \tag{7}$$

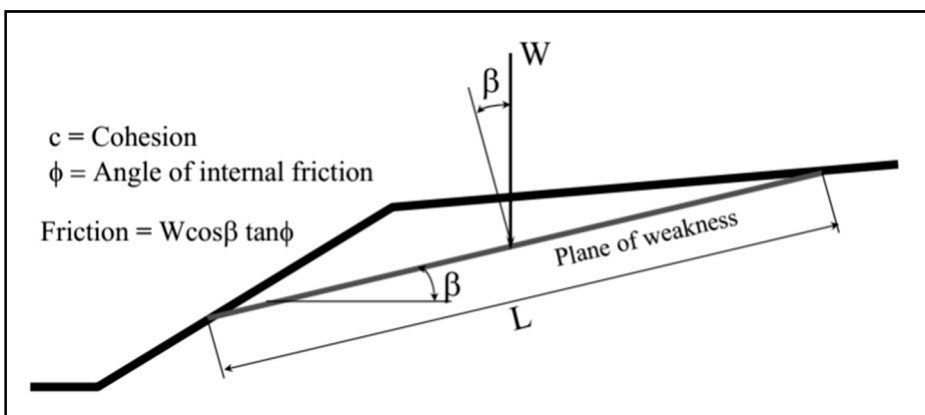

**Figure 4.** Internal Stability of a Soil Slope.

The factor of safety is the ratio of the resisting strength to the force causing failure, which is indicative of the stability of a slope. The forces are in equilibrium when the factor of safety is equal to 1.0. If the forces causing failure are greater than the resisting strength, then the factor of safety is less than 1.0 and the slope will fail. However, a higher factor of safety designates a greater resistance to collapse.

Granular soils (sand and gravel) do not display cohesive behavior under unconfined conditions. A small amount of apparent cohesion may exist due to negative pore water pressure within unsaturated soil particles, although it should not be relied upon for the design of slopes. Additionally, the high permeability of granular soils effectively prevents excess pore water pressure. Conversely, considerable cohesive strength is found in fine grained soils (clay and silt) due to inherent negative pore water pressure that leads to increased effective stress. Since pore water pressure increases in a soil mass with low permeability, the analysis for fine grained soils may be performed using an undrained Mohr–Coulomb failure envelope with an internal friction angle equal to zero [34].

According to the Mohr–Coulomb failure criteria theory, a material fails because of a combination of normal stress and shear stress. The combination of stresses creates a more critical limiting state than would exist if the principal stresses were acting individually. This concept is illustrated by the Mohr–Coulomb failure envelope in Figure 5 [34]. Circle A is indicative of a safe stress state since it is plotted below the failure line. Conversely, a critical stress combination is evident for Circle B, which is tangential to the failure envelope.

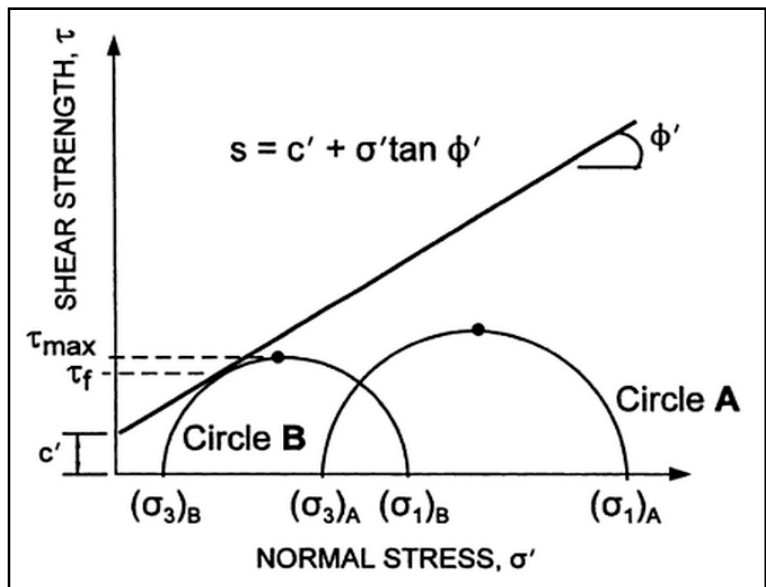

**Figure 5.** Mohr–Coulomb Failure Envelope for Shear Strength of Soils [34].

The shear strength in $N/m^2$ ($\tau$) of an undrained saturated soil is controlled by the material properties and effective stress conditions [35]:

$$\tau = c' + [(\sigma_n - \mu_w) \times tan\phi'] \tag{8}$$

where:

$c'$ = effective cohesion of the soil, $N/m^2$.
$\sigma_n$ = normal stress, $N/m^2$.
$\mu_w$ = pore water pressure, $N/m^2$.
$\phi'$ = effective internal friction angle of the soil, degrees.

*4.1. Bishop Method*

To define the limit-equilibrium conditions, Bishop investigated the use of the slip circle in the stability analysis of slopes as per Figure 6 [36]. The circular failure analysis is performed on a cross-section by dividing it into vertical slices, resolving forces on each slice to calculate the factor of safety, and summing all slice results over the entire slope to obtain an overall factor of safety. It is

typically used for a quick analysis of a simple slope composed of unconsolidated materials. The following equation is used to calculate the factor of safety (*FS*):

$$FS = \frac{\Sigma\{[(c' \times b) + [W + (P \times cos\beta) - (\mu w \times b \times sec\alpha)] \times tan\phi']/m\alpha\}}{\Sigma(W \times sin\alpha) - [(\Sigma MP)/R]} \tag{9}$$

where:

$c'$ = effective cohesion of the soil, N/m$^2$.
$b$ = width of the slice, m.
$W$ = weight of the slice, N.
$P$ = total normal force on the base of the slice, N.
$\beta$ = slope angle, degrees.
$\mu_w$ = pore water pressure, N/m$^2$.
$\alpha$ = inclination angle of the base of the slice, degrees.
$\phi'$ = effective internal friction angle of the soil, degrees.
$m_\alpha$ = $cos\alpha$ + [(sin$\alpha$ × tan$\phi'$)/FS].
$M_P$ = moment about the center of the circle produced by $P$, N·m.
$R$ = radius of the circle, m.

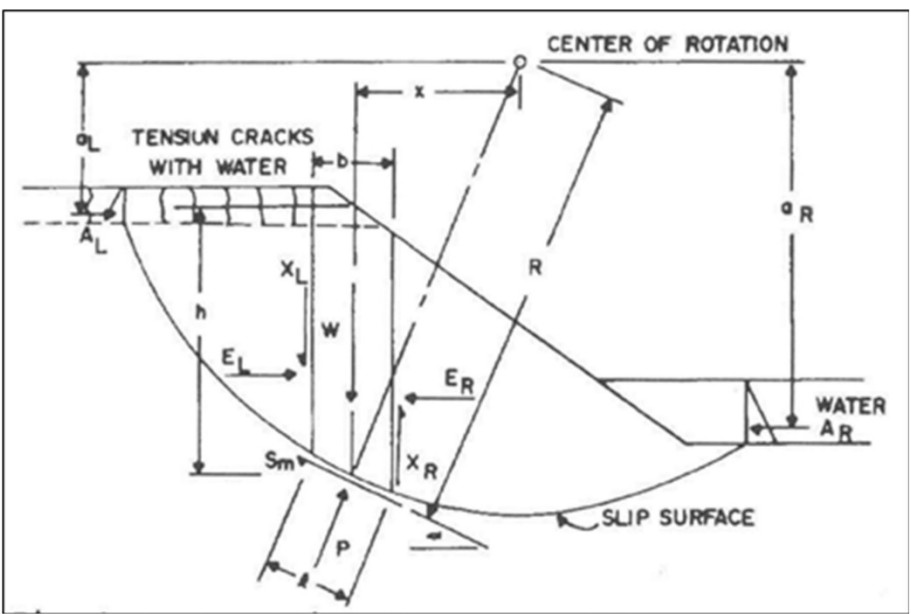

**Figure 6.** Vertical Slice for Internal Stability Analysis [37].

An iterative trial and error procedure is required to solve the equation since the factor of safety appears on both sides. Factors of safety calculated by the Bishop method are comparable with those calculated using other methods. Investigators have shown that it is typically within 5 percent of other solutions [36]. However, since horizontal forces are not satisfied, it is not recommended for seismic analysis where additional horizontal forces are applied. The Bishop method assumes no interslice shear force but Janbu's simplification method accounts for interslice shear forces.

*4.2. Spencer Method*

Spencer developed another method of analysis for embankment stability by assuming parallel interslice forces as per Figure 7 [38]. It can be used for both circular and non-circular failure surfaces in simple structures and can verify a Bishop analysis. Also, whereas the Bishop method only satisfies the

vertical force and moment equilibrium, the Spencer limit-equilibrium technique balances the horizontal force, vertical force, and moment equilibrium. An iterative trial and error procedure is repeated in which values for the factor of safety and side force inclination are assumed until all conditions of force and moment equilibrium are satisfied for each slice. The following equations are used to calculate the uphill and downhill interslice forces as well as the net system moment:

$$Zu = \frac{[(c/FS) \times b \times sec\alpha] - (\gamma \times sin\alpha) + \left\{(tan\phi/FS) \times [(\gamma \times cos\alpha) - (\mu \times b \times sec\alpha)]\right\}}{cos(\alpha - \delta u) \times \left\{1 + [(tan\phi/FS) \times tan(\alpha - \delta u)]\right\}} \qquad (10)$$

$$Z_d = \frac{cos(\alpha - \delta d) \times \left\{1 + [(tan\phi/FS) \times tan(\alpha - \delta d)]\right\}}{cos(\alpha - \delta u) \times \left\{1 + [(tan\phi/FS) \times tan(\alpha - \delta u)]\right\}} \qquad (11)$$

$$M_n = \Sigma\{0.5 \times Z_d \times [(sin\delta_d \times (b_i + b_j)) - [cos\delta_d \times ((b_i \times tan\alpha_i) + (b_j \times tan\alpha_j))]]\} \qquad (12)$$

where:

$Z_u$ = interslice force on the upslope side, N.
$Z_d$ = interslice force on the downslope side, N.
$M_n$ = net system moment, N·m.
$c$ = cohesion of the soil, N/m$^2$.
$FS$ = factor of safety.
$b$ = width of the slice, m.
$\alpha$ = inclination angle of the base of the slice, degrees.
$\gamma$ = unit weight of the soil, N/m$^3$.
$\phi$ = internal friction angle of the soil, degrees.
$\mu$ = pore pressure, N/m$^2$.
$\delta$ = angle of interslice force, degrees.
$u,d$ = upslope or downslope side of the slice.
$i,j$ = slice number.

The interslice force can vary between slices and the value of the angle is represented by the following function:

$$tan\delta_i = (k_i \times tan\theta) \qquad (13)$$

where:

$\delta_i$ = angle of interslice force on the upslope side, degrees.
$k_i$ = 1 (linearly reduced to 0 over the last 20 percent of slices).
$\theta$ = constant angle (Spencer's theta), degrees.

Equilibrium is reached for the factor of safety and the value of Spencer's theta for which the resultant moment and the upslope side interslice force on the last slice are equal to zero. Moreover, the following force and moment equations must be satisfied:

$$Z_n(F,\theta) = 0 \qquad (14)$$

$$M_n(F,\theta) = 0 \qquad (15)$$

The Spencer method requires computer software to perform the calculations to satisfy the moment and force equilibrium for every slice. Calculations are also repeated for a number of trial factors of safety and interslice force calculations. This stability analysis method is used when a statically complete solution is desired, and it can be checked using the force equilibrium procedure [39].

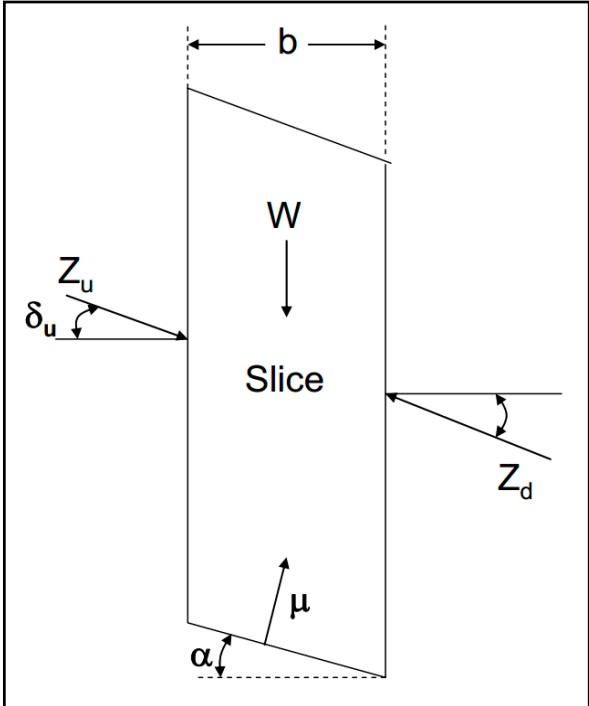

**Figure 7.** Spencer Method of Internal Stability Analysis.

### 4.3. Jewell Method

Jewell investigated the effects of reinforcement on the mechanical behavior of soils [40]. Findings indicated that the state of stress was modified due to the shear generated by the tensile reinforcement. The horizontal forces required to maintain equilibrium are calculated as a gross force as follows:

$$T = (0.5 \times K \times \gamma \times H^2) \tag{16}$$

where:

$T$ = tensile force, N/m.
$K$ = equivalent earth pressure coefficient.
$\gamma$ = unit weight of the soil, N/m$^3$.
$H$ = height of the slope, m.

Additional design equations and charts were also developed by Jewell that allow for determination of the earth pressure coefficient and the length of reinforcement as a function of the slope angle, soil friction angle, and water pressure parameter [41].

### 4.4. Leshchinsky Method

An approach for stability analysis of geosynthetic reinforced steep slopes over firm foundations has also been presented by Leshchinsky and Boedeker based on satisfying limit-equilibrium requirements [42]. A homogeneous soil mass with no pore pressure contained between the slope and slip surfaces is shown in Figure 8. The slip surface is often taken as a log spiral extending between the crest and toe. For the reinforcement capacity to meet the required design tensile resistance, it must be embedded beyond the slip surface. To achieve equilibrium, the pullout resistance should equal its design tensile resistance:

$$t_j = 2 \times k \times tan\phi \times \sigma \times L_{ej} \tag{17}$$

where:

$t_j$ = pullout resistance per unit width of geosynthetic sheet ($j$), N/m.

$k$ = coefficient of friction at the soil-geosynthetic interface.

$\phi$ = internal friction angle of the soil, degrees.

$\sigma$ = average normal stress, N/m$^2$.

$L_{ej}$ = embedment length of geosynthetic sheet ($j$) beyond the slip surface, m.

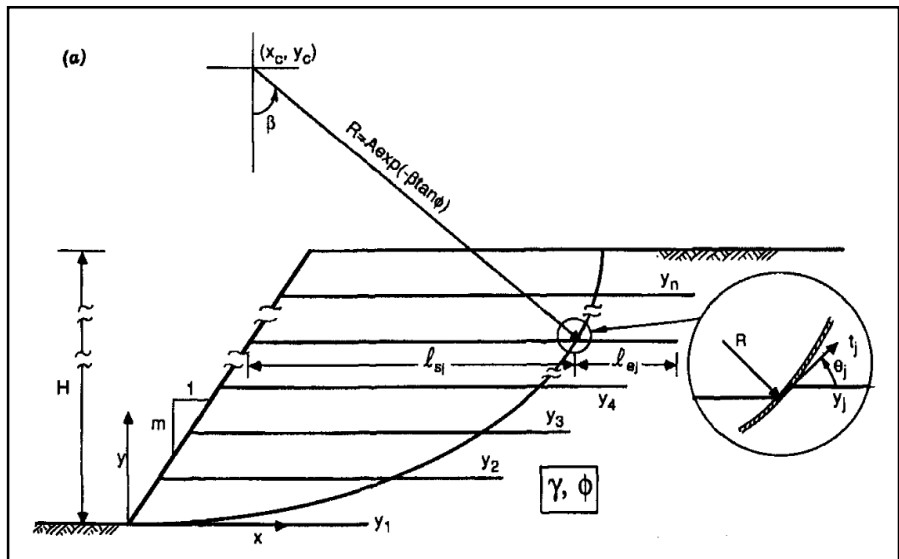

**Figure 8.** Leshchinsky Method of Stability Analysis [42].

### 4.5. Federal Highway Administration Method

Global safety factors and performance limits are traditionally used to establish the adequacy of earthwork and structural foundation design features in the United States. In the FHWA design guidelines, the maximum tension that acts on each level of the reinforcement is determined by considering the necessary tensile strength of reinforcement and soil shear resistance to reach local equilibrium [42]. Although various methods of vertical slices have significantly refined the design technique, the traditional limit-equilibrium approach is purely static as it assumes that soil at failure obeys the perfectly plastic Mohr-Coulomb criterion [43]. This type of procedure is limited, however, as it does not consider the reinforcement stiffness and compaction effects in the analysis [44]. Some authors have proposed methods based on working stress conditions to overcome the deficiencies of the limit-equilibrium method [45,46]. Others have proposed formulations for estimating geosynthetic reinforcement behavior under pullout efforts, but the current practice is to adopt conservative estimates [47]. See Appendix B for the design example.

## 5. Construction Practices

The construction of geosynthetic reinforced steep slopes is similar to normal embankment construction, as the reinforcement can be easily incorporated into the backfill material [6]. Information from transportation agencies, manufacturers, and engineers was synthesized by the research team to determine current methods.

### 5.1. Construction Sequence

The following step-by-step procedure is recommended by the FHWA for the construction sequence of geosynthetic reinforced steep slopes.

During site preparation, contractors clear and grub the site and remove all slide debris. A level subgrade is then prepared. The foundation must also be inspected and drainage features should be placed as required [48].

The reinforcement is then placed with the principal strength direction perpendicular to the face of the slope, pulled taut, and secured with retaining pins to prevent movement during fill placement. The geosynthetic materials should extend back from the slope face to the specified embedment distance at the elevations shown on the drawings. Adjacent reinforcement must also be butted together side by side without overlap unless specified [1].

Backfill material is placed and compacted without deforming or moving the reinforcement, utilizing lightweight compaction equipment near the slope face to maintain alignment. Maintain a minimum of 15 cm of fill between the reinforcement and the wheels or tracks of construction equipment. Never operate tracked equipment directly upon the reinforcement [49]. Sudden braking and sharp turning should be avoided. Tracked equipment should not turn within the reinforced fill zone to prevent tracks from displacing the fill and damaging the reinforcement. Rubber-tired equipment may operate directly on the reinforcement if the travel is infrequent, equipment travels slow, turning is minimized, and no damage or displacement to the reinforcement is observed. Water content and backfill density must be monitored to control compaction [48]. Fill should be compacted to at least 95 percent of the standard AASHTO T-99 maximum density within 2 percent of optimum moisture [10,50].

### 5.2. Transportation Agency Specifications

In addition to the aforementioned specifications, several state transportation agencies have implemented similar requirements for construction involving geosynthetic reinforcement. Geosynthetics should be delivered to the jobsite in unopened shipping packages labeled with the supplier's name, product name, quantity, and type designation that corresponds to that required by plans [27]. They should also be accompanied by a manufacturer certified copy of test results (ASTM D4595 for geotextile or ASTM D6637 for geogrid), verifying the ultimate strength of the lots from which the rolls were obtained [20,51,52]. Reject all rolls damaged during transport. Rolls should be protected from construction equipment, chemicals, sparks and flames, water, mud, wet cement, epoxy, and like materials [22]. Geosynthetics should also be protected from direct sunlight and temperatures below −20 °C per Mississippi DOT [53]. Additionally, the reinforcement material must be protected from temperatures above 50 °C according to Mississippi DOT or above 60 °C and 70 °C as per Pennsylvania DOT and California DOT, respectively [21,28,53]. Rolls should also be stored elevated from the ground and covered with a waterproof cover.

Remove all existing vegetation and unsuitable soil materials, including deleterious materials and soils, from the grade. Grade should be proof rolled with five passes of a static, smooth drum, or pneumatic tire roller with a minimum contact pressure of about 0.8 MPa as per Pennsylvania DOT [28]. However, according to Florida DOT, a vibratory or sheepsfoot roller of at least 1.7 MPa on the tamper foot should be used [27]. Any soft areas, as determined by the engineer, should be removed and replaced with backfill. Benching the backcut into competent soil is recommended to improve stability.

The geosynthetic should be oriented with the direction of maximum strength perpendicular to the slope face with each layer placed to form a continuous mat. Secondary reinforcement should be placed in continuous strips parallel to the slope face, and the geosynthetic must be secured and pulled taut before placing any fill. West Virginia DOT restricts operation of equipment on geosynthetic materials until 15 cm of loose backfill has been placed, although Pennsylvania DOT further limits this to 20 cm [22,28]. Sudden braking and sharp turning is not allowed. Sheepsfoot or padfoot type compaction equipment is not allowed. Only place the amount of geosynthetic that can be covered in one day and slope the last level of backfill away from the slope face to allow for positive drainage.

If mechanical connectors are required, the splice mechanism must allow a minimum of 95 percent load transfer from piece to piece of geosynthetic per Florida DOT [27]. Only one joint per length of

reinforcement shall be allowed, and the joint shall be made for the full width of the strip by using a similar material with similar strength that uses a connection device supplied or recommended by the manufacturer. Pennsylvania DOT does not allow for any splicing of any geosynthetic [28]. If a geosynthetic is damaged, remove all backfill material from the area plus 1.2 m in all directions beyond the limits of the damage. Patch the damaged area with the same material and overlap the undamaged patch a minimum of 1 m in all directions. Agencies should also consider site specific installation damage testing, especially if relatively coarse, uniformly graded crushed, or otherwise angular aggregate is used as backfill, or if other relatively severe installation conditions are anticipated.

## 6. Performance Measures & Cost Effectiveness

Geosynthetic reinforced steep slopes can yield potential savings in material costs and construction time for new permanent embankments and recurring slope failures. However, an understanding of their performance is required to effectively use this method of mechanically stabilizing earth.

### 6.1. Quality Control

Properties of geosynthetics are classified as general, index, and performance. General properties include the polymer, mass per unit area, thickness, roll dimensions, roll weight, and specific gravity. However, index properties provide qualitative assessment through standard test procedures for utilization in product comparisons, specifications, quality control, and constructability. Index tests include uniaxial mechanical strength, multiaxial rupture strength, durability tests, and hydraulic tests. Finally, performance properties of geosynthetics include the use of soil samples for direct assessment of properties, such as in-soil stress-strain, creep, friction/adhesion, chemical resistance, and filtration. Under direction of the design engineer, performance tests are correlated to index values for use in preselecting geosynthetics for project specifications.

Geosynthetic specifications include general requirements, specific geosynthetic properties, seams and overlaps, placement procedures, repairs, and acceptance and rejection criteria [1]. Specified general requirements include the types of geosynthetics, acceptable polymeric materials, and guidelines for the stability of the materials as well as instructions for storage and handling, roll weight, and dimensions and certification requirements. Physical, index, and performance properties are also included in the general requirements as per the specific project design. Additionally, seams and overlaps are clearly specified in all geosynthetic applications in accordance with construction requirements. Although overlaps may be increased, a minimum overlap of 30 cm is recommended for geotextile applications. However, geosynthetics are connected if overlaps will not work, and the connection material should always consist of polymeric materials that have equal or greater durability than the specified geosynthetic. Placement procedures are detailed on the construction drawings and typically include requirements for grading, ground-clearing, aggregates, lift thickness, and equipment. Repair procedures for damaged sections, acceptance and rejection criteria, and inspection procedures, including sampling and testing requirements, are also included in geosynthetic specifications to ensure successful project completion.

### 6.2. Failure Modes

Koerner and Koerner compiled and analyzed a database of 141 failed geosynthetic reinforced structures, in which there were 34 cases of excessive deformation and 107 cases of actual collapse [54]. Below are the main statistical findings:

- All but one were privately owned (as opposed to publicly financed).
- 72 percent were in North America.
- 49 percent were 4 m to 8 m high.
- 90 percent were geogrid reinforced.
- 81 percent failed in less than 4 years.
- 62 percent used silt or clay backfill in the reinforced soil zone.

- 75 percent had poor to moderate compaction.
- 98 percent were caused by improper design or construction.
- 58 percent were caused by internal or external water (the remaining 42 percent were caused by soil related issues).

The major inadequacies in these failures were a lack of proper drainage procedures and a lack of adequate placement of plumbing within the reinforced soil zone. Another inadequacy was the use of fine grained silt and clay backfill soils along with insufficient placement and compaction during construction. This led to hydraulic pressures being mobilized behind or within the reinforced soil zone, which requires the use of back and base drains so as to dissipate the pressures and properly remove the water out of the front.

Liu et al. also investigated three failures on a geogrid reinforced slope on the approaching road to Chi-Nan University in Nantou, Taiwan [55]. The first slope failure occurred after rainfall infiltrated into the permeable gravel and was impeded by the underlying impermeable clay layer. The shear strength was reduced at the interface of the gravel and clay, initiating a slide when the toe of the slope was excavated to install reinforcement during construction. The second failure was caused by a strong earthquake, which generated stress and created a slide in the vicinity of the clay layer. The third failure occurred when abundant rainfall infiltrated into the reinforced slope during a heavy rainstorm. The infiltration was obstructed by the impermeable clay, producing transient water pressure and inducing slope failure behind the reinforced zone. Lessons learned from the study include carrying out a detailed site investigation, selecting permeable materials as backfill, installing drainage systems appropriately, and combining the design of a reinforced slope with other types of retaining structures to improve the system global stability.

## 7. Conclusions & Recommendations

The purpose of the study was to provide a review of the current technology on geosynthetic reinforced steep slopes in the United States. The following conclusions were made based on a comprehensive review of technical reports, journal articles, transportation agency specifications, and case studies, as well as surveys and interviews of academic and industry professionals:

1.  Reinforced slopes are a form of mechanically stabilized earth that incorporate planar reinforcing elements for the construction of sloped structures with inclinations less than 70°. To provide tensile resistance and stability, geosynthetic reinforcement has been employed for repairing failed slopes, constructing new embankments, and widening existing embankments.
2.  According to the survey, following the FHWA design guidelines is the most advocated approach for designing reinforced slopes. Other common design methods have been developed by Jewell, Leshchinsky, and Eurocode. Internal and external stability are considered, including rotational, sliding, bearing, and lateral failure.
3.  Geogrids and geotextiles manufactured from polyester, polypropylene, or polyethylene are commonly used for reinforcement. The material must be resistant to all naturally occurring alkaline and acidic soil conditions, resistant to heat, ultraviolet light, and to attack by bacteria and fungi in the soil.
4.  An adequate subsurface investigation should be performed for the existing foundation, as well as behind and in front of the structure, to assess overall performance behavior. To facilitate compaction, a high quality fill meeting gradation, shear strength, and internal friction angle requirements are recommended for the embankment soil.
5.  The reinforcement is placed with the principal strength direction perpendicular to the face of the slope, pulled taut, and secured with retaining pins to prevent movement during fill placement. Backfill material is placed and compacted without deforming the reinforcement, utilizing lightweight compaction equipment near the slope face to maintain alignment.

6.  Performance monitoring programs are recommended for cases in which new features or materials have been incorporated in the design, post construction settlements are anticipated, or where degradation/corrosion rates of reinforcements are to be monitored.

7.  Geosynthetics can be degraded by a combination of environmental mechanisms. However, none of the documented failures were due to inadequate reinforcement. All failures within the reviewed body of literature were due to improper design in the area of surface and internal water removal or the use of fine grained silt and clay backfill soils.

8.  Many site specific characteristics contribute to the overall cost, including cut-fill requirements, slope size and type, existing soil type, available backfill materials, facing finish, and application. The approximate costs of the principal components are as follows: reinforcement 45–65 percent, backfill 30–45 percent, and face treatment 5–10 percent.

The following recommendations were made based on the knowledge and experience gained during the study:

1.  There is a need to use the specification and construction checklists, as well as implementing a comprehensive performance monitoring program, in order to compare the observed behavior and the intended design.

2.  The FHWA design guidelines should be followed. In addition, there is a need to follow the existing topography, soil properties, and subsurface conditions in evaluating the construction site. The investigation should also seek to determine the backfill materials as well as availability of the required type of reinforced fill.

3.  It is also recommended that the polyester, polypropylene, or polyethylene's constructed geotextile or geogrid should be used in reinforcing deep slopes. The materials should have characteristics such as heat resistance, ultraviolet resistance, and fungi and bacteria attack resistance, as well as resistance to alkaline and acidic soil conditions.

4.  There is also a need to use free draining backfill having the recommended gradation limits of AASHTO T-27 in the reinforced volume. Moreover, there should be a plasticity index of less than 20 and a pH level of 5–10 in addition to the reinforced fill being reasonably free from organic or other such deleterious materials. When developing design calculations, factors such as soils density, cohesion, and internal friction angle must be determined and taken into consideration.

5.  It is also recommended that the surface water runoff must be collected above the slope that has been reinforced prior to channeling the water to slope's base. The subsurface water drainage features must also be designed in order to address the challenges of filtration, flow rate, placement, and outlet details.

6.  Facing elements should also be designed appropriately such that if the slope facing is designed to prevent soil erosion, the face's reinforcement should turn up and return into the next reinforcement layer's embankment below it. The other most commonly used facing elements include different meshes made up of steel or polymers, which make it possible for the face to be vegetated after it has been constructed.

**Author Contributions:** Y.-J.K. and B.H.Y. conceived and designed the survey and case study; A.R.K. and Y.-J.K. performed the survey; A.R.K. and J.W. analyzed the survey data; B.-Y.C. and B.H.Y. contributed reagents/materials/analysis tools; Y.-J.K., A.R.K., and J.W. wrote the paper.

**Funding:** This research was supported by Research Grant No. 0-6792-1 from Texas Department of Transportation.

**Acknowledgments:** The authors are grateful for the support and guidance of Kevin Pete, Project Manager of Research and Technology Implementation at the Texas Department of Transportation. In addition, the authors thank Marie Fisk, Sean Yoon and John Delphia for their valuable comments during the project. Compliance with ethical standards.

**Conflicts of Interest:** The authors declare no conflict of interest.

## Appendix A

### Case Study #1: Yeager Airport Runway

**Location:** Charleston, West Virginia
**Owner:** Yeager Airport
**Engineer:** Triad Engineering
**Contractor:** Cast & Baker
**Purpose:** Support Structure for Airport Runway

**Geosynthetic Material:**

- Reinforcement: Polyester Geogrid (12.9 kN/m & 18.6 kN/m)
- Facing: Polypropylene Mesh
- Drainage: Polypropylene Geotextile

**Foundation & Embankment Soil:** The onsite geomorphology consisted of weathered sandstone underlain by sandstone and some shale. Testing of the weathered sandstone soil showed it to have a maximum dry density of 2034 kg/m$^3$ and a peak friction angle of 39°. The compressive strength of the rock foundation varied from 30 MPa to 95 MPa.

**Slope Height:** 73.8 m
**Slope Angle:** 1H:1V

**Design Method:** Limit Equilibrium Analysis

**Construction Sequence:**

- The existing ground was excavated to the required level to provide a stable platform for the reinforced slope.
- The geogrids were installed as horizontal reinforcing elements into the slope in conjunction with the backfill material.
- Embedment lengths of the geogrid were on the order of 59.4 m in length.
- A drainage composite was installed along the back of the excavation to intercept and drain seepage water from the existing mountain side away from the reinforced mass.
- Mesh was installed on the face of the slope at 0.6 m vertical intervals, with 0.9 m embedded into the slope face and 0.76 m down the face for facial stability and erosion protection.

**Performance:** The reinforced slope was successfully completed and is performing as expected. Geogrids provided the high strengths required for a structure of this size, and the mesh allowed for facing stability and quick germination of surficial vegetation for improved stability.

**Cost Effectiveness:** Construction options for extending the runway past the existing hillside included evaluation of bridge structures, retaining walls, and reinforced slopes. Engineering evaluation indicated the reinforced slope provided the most cost effective and easiest constructed option of the structures considered.

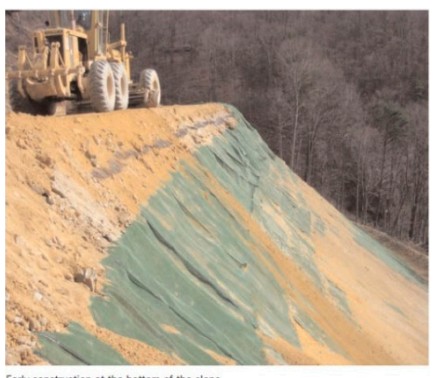

Early construction at the bottom of the slope.

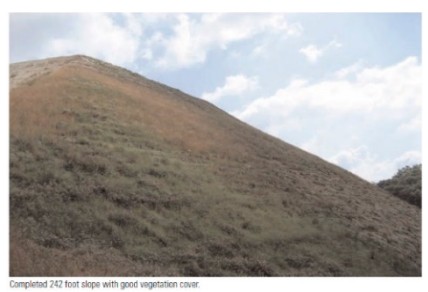

Completed 242 foot slope with good vegetation cover.

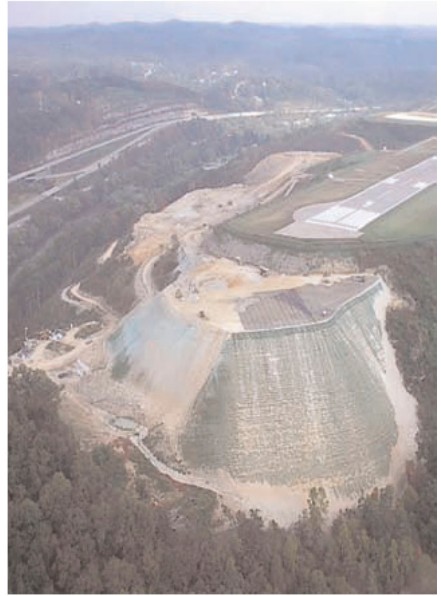

Aerial photo of slope during construction, approximately 80% complete.

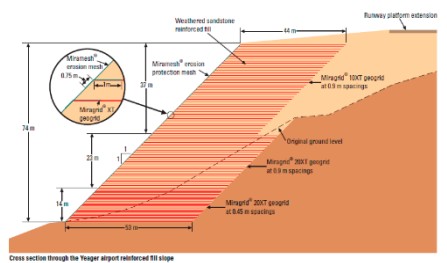

Cross section through the Yeager airport reinforced fill slope

**Case Study #2: Reinforced Slope Failure & Reconstruction**

**Location:** Hondo Valley, New Mexico
**Owner:** New Mexico Department of Transportation
**Engineer:** Kleinfelder
**Contractor:** Sierra Blanca Constructors
**Purpose:** Support Structure for Roadway

**Geosynthetic Material:**

- Reinforcement: Geogrid

**Slope Height: 7.6** m
**Slope Angle:** 1H:2V

**Construction Specifications:** New Mexico DOT

**Construction Sequence:**

- The geotechnical investigation consisted of drilling two soil borings adjacent to the deepest parts of the roadway fill. The investigation indicated that the geology in the area was terrace deposits and alluvium overlying Yeso formation mudstone.
- Groundwater was not encountered in the borings drilled at the site, and no signs of slope instability were noted during the site reconnaissance.
- A major geotechnical slope failure occurred a year after construction due to movement associated with severe rain storms. The project team excavated a test trench through the roadway to evaluate the conditions at the cracked area, and noted vertical cracks observed at the surface were about 5.1 cm deep and were located directly behind the top layer of geogrid.
- Additional field exploration activities began as the team drilled seven additional borings below the roadway. The borings were used to establish a detailed stratigraphic log of the subsurface conditions at the GRSS and to collect soil samples for laboratory testing.
- The maximum cumulative displacement measured at the top of the inclinometers was approximately 8.9 cm, and a clear failure surface was apparent just beneath the bottom geogrid.
- The analysis concluded that the distress was the result of peak flows and saturation during heavy and prolonged rainfall that developed excess water pressure at the rear of the slope.
- The contractor excavated about 6.1 m long segments to remove unsuitable material below the embankment. The back and sides of the excavation were lined with filter fabric to reduce the potential for piping of fines from the native materials into the rock fill.
- On the back of the excavation, geocomposite drain fabric was placed that extended vertically from just below the existing pavement down to the smaller rock layer.

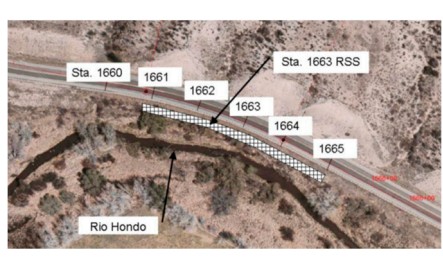

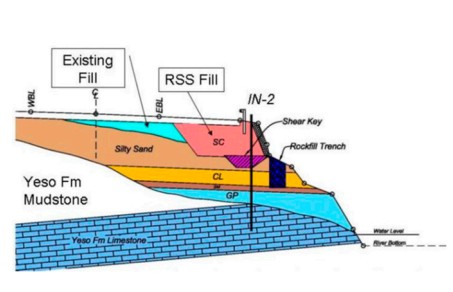

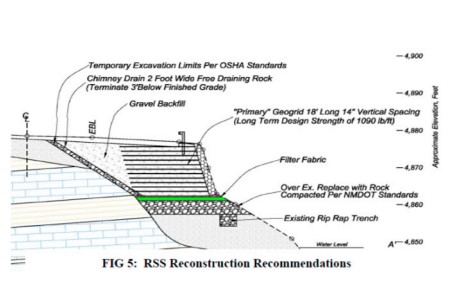

FIG 5: RSS Reconstruction Recommendations

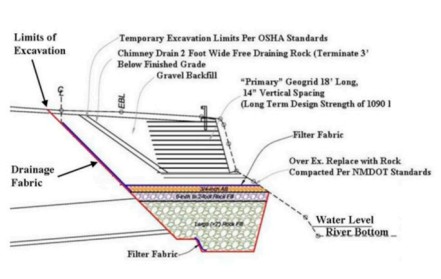

**Appendix B**

**FHWA Design Guidelines with Chart Solution**

The following example provides a step by step procedure for performing a GRSS preliminary design with a chart solution in order to evaluate the feasibility of specific project parameters. **Define**

**Slope Geometry, Loading, and Performance**

The first step was to establish the geometric, loading, and performance requirements for the design. Factors of safety recommended by the FHWA were used for the analysis.

Geometric and Load Requirements:

> Slope Height (H) = 6.1 m
> Slope Angle ($\beta$) = 70°
> Surcharge Load (q) = 12 kN/m$^2$
> Crest Width (A) = 6.1 m

Performance Requirements:

> Internal Stability: FS = 1.5
> External Stability: FS = 1.5

**Define Engineering Properties of Soils**

The parameters listed below were determined for the foundation, retained, and reinforced soils.

Foundation and Retained Soils:

> Internal Friction Angle ($\phi_u'$) = 34°
> Cohesion ($c_u'$) = 0
> Density ($\gamma_u$) = 19.6 kN/m$^3$

Reinforced Soil:

> Internal Friction Angle ($\phi_r'$) = 34°
> Cohesion ($c_r'$) = 0
> Density ($\gamma_r$) = 19.6 kN/m$^3$
> Depth of Water Table ($d_w$) = 5 ft

**Determine the Geogrid Force Coefficient and Total Design Tension**

> $\phi_f = [\tan^{-1} \times (\tan\phi_r/FS)] = [\tan^{-1} \times (\tan34°/1.5)] = 24.2°$
> K = 0.34 as per reference 10
> $T_{S-MAX} = [0.5 \times K \times \gamma_r \times (H')^2] = \{0.5 \times 0.34 \times 19.6 \times [6.1 + (12/19.6)]^2\} = 150.1$ kN/m

**Determine the Reinforcement Vertical Spacing and Design Tension per Reinforcement Layer**
$S_v$ = 0.4 m

> $T_{MAX} = [(T_{S-MAX} \times S_v)/H)] \times (RF_{ID} \times RF_{CR} \times RF_D)$
> $\quad\quad = [(150.1 \times 0.4)/6.1)] \times (1.2 \times 3.0 \times 1.25)$
> $\quad\quad = 44.3$ kN/m

**Determine Length of Reinforcement**

> L/H′ = 0.8 as per reference 10

$$L = (L/H') \times H' = 0.8 \times [6.1 + (12/19.6)] = 5.37 \text{ m}$$

**Jewell Method**

The following example provides a step by step procedure for performing a GRSS preliminary design using the Jewell Method in order to evaluate the feasibility of specific project parameters.

Consider a slope with the following parameters:

| | |
|---|---|
| $H = 6.1$ m | (height) |
| $\beta = 70°$ | (slope angle) |
| $q = 10.1$ kN/m$^2$ | (surcharge load) |
| $c = 0$ | (soil cohesion) |
| $\phi = 32°$ | (soil internal friction angle) |
| $\gamma = 20$ kN/m$^3$ | (soil unit weight) |
| $FS_{design} = 1.30$ | (design factor of safety) |
| $FS_{grid} = 2.75$ | (geogrid factor of safety) |
| $T_{ult} = 65.7$ kN/m | (geogrid tensile strength) |
| $r_u = 0.25$ | (pore water pressure coefficient) |

(1)  Calculate the allowable tensile strength ($T_{allow}$) and design tensile strength (P).

$$T_{allow} = (T_{ult}/FS_{grid}) = (65.7/2.75) = 23.9 \text{ kN/m}$$
$$P = (T_{allow}/FS_{design}) = (23.9/1.30) = 18.4 \text{ kN/m}$$

(2)  Based on the pore water pressure coefficient, determine the coefficient of earth pressure (K) and the ratios of reinforcement length to embankment height $(L/H)_{overall}$ and $(L/H)_{sliding}$ using the charts in reference 10, then calculate the reinforcement length.

$$K = 0.30$$
$$(L/H)_{overall} = 0.65$$
$$(L/H)_{sliding} = 0.60$$
$$H' = [H + (q/\gamma)] = [6.1 + (10.1/20)] = 6.6 \text{ m}$$
$$L = [H' \times (L/H)_{overall}] = (6.6 \times 0.65) = 4.3 \text{ m}$$

(3)  Define the spacing constant (Q) for the slope in terms of the minimum spacing (v) to be used.

$$v = 0.2 \text{ m}$$
$$Q = [P/(K \times \gamma \times v)] = [18.4/(0.30 \times 20 \times 0.2)] = 15.33 \text{ m},$$

(4)  Define the zones for reinforcement layers spaced equally at $v_1, v_2, v_3, \ldots v_n$.

| i | Spacing ($Sv_i$) | Depth ($Z_i$) | Thickness ($s_i$) |
|---|---|---|---|
| 1 | $1v = 0.2$ m | $Q = 15.33$ m | - |
| 2 | $2v = 0.4$ m | $Q/2 = 7.67$ m | $7.67 - 5.11 = 2.56$ m |
| 3 | $3v = 0.6$ m | $Q/3 = 5.11$ m | $5.11 - 3.83 = 1.28$ m |
| 4 | $4v = 0.8$ m | $Q/4 = 3.83$ m | |

(5)  Calculate the number and position of the required reinforcement layers. The number of grids in a zone (N) is rounded down to the nearest whole number.

| i | $s_i'/Sv_i$ | Number of Grids ($N_i$) | Remaining Thickness ($R_i = s_{i-1} - (Sv_i \times N_i)$) | $s_{i+1}' = s_{i+1} + R_i$ |
|---|---|---|---|---|
| 0 | | 1 | $R_0 = $ base | |
| 1 | 2.56 m/0.4 m = 6.4 | 6 | $R_1 = 2.56 - (0.4 \times 6) = 0.16$ m | $s_2' = 1.28 + 0.16 = 1.44$ m |
| 2 | 1.44 m/0.6 m = 2.4 | 2 | $R_2 = 1.44 - (0.6 \times 2) = 0.24$ m | $s_3' = 3.83 + 0.24 = 4.07$ m |
| 3 | 4.07 m/0.8 m = 5.09 | 5 | $R_3 = 4.07 - (0.8 \times 5) = 0.07$ m | |

If the top layer of reinforcement is more than 0.6 m below the slope crest, it is prudent to add an additional layer. Therefore, although $N_{total} = 14$, a 15th reinforcement layer spaced 0.6 m near the crest of the slope should be added.

(6)　　Calculate the gross horizontal force for equilibrium and check the geogrid tensile force.

$$T = [0.5 \times K \times \gamma \times (H')^2] = [0.5 \times 0.30 \times 20 \times (6.6)^2] = 130.7 \text{ kN/m}$$
$$T/N_{total} = 130.7/15 = 8.71 \text{ kN/m}$$
$$T/N_{total} \leq P$$

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
