# Peer review of "Geosynthetic Reinforced Steep Slopes: Current Technology in the United States"

_applsci, doi:10.3390/app9102008_

Round 1

Reviewer 1 Report

The article cover the topic of the Geosynthetic Reinforced Steep Slopes.
The authors focus in detail on the techniques used in the USA.
This work is good and the presented vievs are supported by
determined and accurately performed experimental procedures.

Some minor revisions recommended as below:

1. I suggest the following keywords in this order: Reinforcement, Geosynthetics, Geogrid, Soil, Slope
2. Line 56 - 'In lieu' - it should be improved
3. Authors should determine the abbreviations: 'DOT';, 'PET, PE, PP' (Table 1)
4. Table 2 - 'sieve size, percent passing' - define unit
5. Line 148 - What kind of soil could be named 'soft soil'?
6. Parameters without unit, ex. Nc, FS (page 5) Epsylon (Strain) etc. should be determined as dimensionless [-].
7. Figure 4 - In the picture, friction angle of the soil should be drawn as 'phi'
8. I recommend to add and describe 'Janbu method' (paragraph Design methods)
9. In my opinion, the article should be supplemented with the material aspect (types of reinforcements, materials,
properties, application, strength, deformability, corrosion resistance).

Author Response

April 26, 2019

Sustainability Journal,

Manuscript number: applsci-490070, “Geosynthetic Reinforced Steep Slopes: Current Technology in the United States”

Dear Reviewers,

Thank you very much for your efforts in helping us to make the manuscript mentioned above into a better paper. In addition, we would like to thank the reviewer for his valuable comments on the manuscript. The authors revised the manuscript with all the comments and all the issues have been adequately addressed. Please check the attached responses.

Sincerely,

Dr. YooJae Kim

Associate Professor

Department of Engineering Technology

Texas State  University

San Marcos, TX  78666

Explanation on comments from reviewer 1:

1) I suggest the following keywords in this order: Reinforcement, Geosynthetics, Geogrid, Soil, Slope.

® It has been changed following your comment.

2) Line 56 - 'In lieu' - it should be improved.

® It has been changed to “Instead of importing soil to reconstruct the slopes, the slide debris was salvaged and reused with the addition of geosynthetic reinforcement, resulting in reduced costs.”

3) Authors should determine the abbreviations: 'DOT'; 'PET, PE, PP' (Table 1)

® It has been added DOT full name on line 100, and inserted PET, PE, PP in the Table 1.

4) Table 2 - 'sieve size, percent passing' - define unit.

® It has been added the units in the Table 3.

5) Line 148 - What kind of soil could be named 'soft soil'?

® I appreciate your comment on that. It has been changed to weak soil. Weak soil is more appropriate to use in the sentence. Also, the kinds of soft soil are varies on the locations in US. Normally, Gumbo clay, Boston blue clay, and Bangkok clay have been called as soft soil.

6) Parameters without unit, ex. Nc, FS (page 5) Epsylon (Strain) etc. should be determined as dimensionless [-].

® Yes, they should. I have put the m/m unit for strain and for the rest of them, I have just left them as is, since the factor or coefficients would be the dimensionless.

7) Figure 4 - In the picture, friction angle of the soil should be drawn as 'phi' 
® It has been changed as your comment.

8) I recommend to add and describe 'Janbu method' (paragraph Design methods)

® Thank you very much for your comment. I have mentioned it in a comparison between it and the Bishop Method.

9) In my opinion, the article should be supplemented with the material aspect (types of reinforcements, materials, properties, application, strength, deformability, corrosion resistance).

® Table 2 has been added on the manuscript for Geosynthetic Types, Properties, and Test Methods.

Reviewer 2 Report

The article interesting, but most of it is devoted to things already known. The most important part, in my opinion, is given in the appendices on 'case study'. Maybe on their basis it would be possible to reveal the novelty and significance of the study.

Besides, from point of geotechnical view, the theoretical  part of manuscript can be shorter with citation of calculations methods, standards or some comparisions of used methods.
But the part of two case study should be made more detailed analysis with calculations and modeling, and may be some algoritm of decision.
I think, that should be changed structure of the manuscript.

The case history I think are interesting and can help in the futures analysis of reinforced steep slopes.

Author Response

April 26, 2019

Sustainability Journal,

Manuscript number: applsci-490070, “Geosynthetic Reinforced Steep Slopes: Current Technology in the United States”

Dear Reviewers,

Thank you very much for your efforts in helping us to make the manuscript mentioned above into a better paper. In addition, we would like to thank the reviewer for his valuable comments on the manuscript. The authors revised the manuscript with all the comments and all the issues have been adequately addressed. Please check the attached responses.

Sincerely,

Dr. YooJae Kim

Associate Professor

Department of Engineering Technology

Texas State  University

San Marcos, TX  78666

Explanation on comments from reviewer 2:

The article interesting, but most of it is devoted to things already known. The most important part, in my opinion, is given in the appendices on 'case study'. Maybe on their basis it would be possible to reveal the novelty and significance of the study.

Besides, from point of geotechnical view, the theoretical part of manuscript can be shorter with citation of calculations methods, standards or some comparisons of used methods. 
But the part of two case study should be made more detailed analysis with calculations and modeling, and may be some algorithm of decision.
I think, that should be changed structure of the manuscript.

The case history I think are interesting and can help in the futures analysis of reinforced steep slopes.

® I appreciate your comments. However, if I were to follow all the suggestions listed above, the paper must be started again. So, I have added the parts regarding the design examples in the appendix.

Reviewer 3 Report

Please add the explanations for the abbreviations for PET, PE, PP, DOT etc.

You give the reference for ASTM D4355 but not directly behind. Is this intentionally?

Is the low frequency of use of the Eurocode due to the origin of the respondents?

Author Response

April 26, 2019

Sustainability Journal,

Manuscript number: applsci-490070, “Geosynthetic Reinforced Steep Slopes: Current Technology in the United States”

Dear Reviewers,

Thank you very much for your efforts in helping us to make the manuscript mentioned above into a better paper. In addition, we would like to thank the reviewer for his valuable comments on the manuscript. The authors revised the manuscript with all the comments and all the issues have been adequately addressed. Please check the attached responses.

Sincerely,

Dr. YooJae Kim

Associate Professor

Department of Engineering Technology

Texas State  University

San Marcos, TX  78666

Explanation on comments from reviewer 3:

1) Please add the explanations for the abbreviations for PET, PE, PP, DOT etc.

® It has been added DOT full name on line 100, and inserted PET, PE, PP in the Table 1

2) You give the reference for ASTM D4355 but not directly behind. Is this intentionally?

® Yes, the reference origin was from a report that was published in 2010 and 2011. That’s the reason we put the 2008 ASTM manual.

3) Is the low frequency of use of the Eurocode due to the origin of the respondents?

® Yes, it true. The survey was sent to 393 recipients in the USA, and responses were received from 52 for a response rate of 13 percent.  Of those who responded, 43 have experience designing or constructing GRSS.  Most of the respondents recommend the use of the FHWA method for the design of GRSS. Eighty-eight percent of the respondents recommended the use of geogrid or a combination of geogrid and geotextile for slope reinforcement, while the others recommended independent use of geotextile. 

Round 2

Reviewer 2 Report

The opinion of authors are acceptable , but is some
remarks:

Figures should be better quality.

The example, which authors added, should be in the SI system, like is used in the main text of the manuscript. (The International System of Units (SI) is used in the main text (kN, m, ......) but this system does not used in the example. I think, that should be unified system.)

Author Response

April 30th, 2019

Sustainability Journal,

Manuscript number: applsci-490070, “Geosynthetic Reinforced Steep Slopes: Current Technology in the United States”

Dear Reviewers,

Thank you very much for your efforts in helping us to make the manuscript mentioned above into a better paper. In addition, we would like to thank the reviewer for his valuable comments on the manuscript. The authors revised the manuscript with all the comments and all the issues have been adequately addressed. Please check the attached responses.

Sincerely,

Dr. YooJae Kim

Associate Professor

Department of Engineering Technology

Texas State  University

San Marcos, TX  78666

Explanation on comments from reviewer 2:

1. Figures should be better quality.

® I have increased the resolutions, but the figures still the same.

The example, which authors added, should be in the SI system, like is used in the main text of the manuscript. (The International System of Units (SI) is used in the main text (kN, m, ......) but this system does not used in the example. I think, that should be unified system.)

® I appreciate your comment on that. Both the appendix and case studies have been changed to the International System of Units (SI).
